# Translational Proteomic Approach for Cholangiocarcinoma Biomarker Discovery, Validation, and Multiplex Assay Development: A Pilot Study

**DOI:** 10.3390/molecules27185904

**Published:** 2022-09-11

**Authors:** Kamolwan Watcharatanyatip, Somchai Chutipongtanate, Daranee Chokchaichamnankit, Churat Weeraphan, Kanokwan Mingkwan, Virat Luevisadpibul, David S. Newburg, Ardythe L. Morrow, Jisnuson Svasti, Chantragan Srisomsap

**Affiliations:** 1Laboratory of Biochemistry, Chulabhorn Research Institute, Bangkok 10210, Thailand; 2Pediatric Translational Research Unit, Department of Pediatrics, Faculty of Medicine Ramathibodi Hospital, Mahidol University, Bangkok 10400, Thailand; 3Center for Population Health Science and Analytics, Department of Environmental and Public Health Sciences, University of Cincinnati College of Medicine, Cincinnati, OH 45267, USA; 4Department of Molecular Biotechnology and Bioinformatics, Faculty of Science, Prince of Songkla University, Songkla 90110, Thailand; 5Division of Surgery, Sapphasitthiprasong Hospital, Ubon Ratchathani 34000, Thailand; 6Division of Information and Technology, Ubonrak Thonburi Hospital, Ubon Ratchathani 34000, Thailand; 7Division of Infectious Diseases, Department of Pediatrics, Cincinnati Children’s Hospital Medical Center, University of Cincinnati College of Medicine, Cincinnati, OH 45267, USA; 8Applied Biological Sciences Program, Chulabhorn Graduate Institute, Bangkok 10210, Thailand

**Keywords:** biomarker, cholangiocarcinoma, immunoassay, machine learning, multiplex assay, plasma proteomics, translational research

## Abstract

Cholangiocarcinoma (CCA) is a highly lethal disease because most patients are asymptomatic until they progress to advanced stages. Current CCA diagnosis relies on clinical imaging tests and tissue biopsy, while specific CCA biomarkers are still lacking. This study employed a translational proteomic approach for the discovery, validation, and development of a multiplex CCA biomarker assay. In the discovery phase, label-free proteomic quantitation was performed on nine pooled plasma specimens derived from nine CCA patients, nine disease controls (DC), and nine normal individuals. Seven proteins (S100A9, AACT, AFM, and TAOK3 from proteomic analysis, and NGAL, PSMA3, and AMBP from previous literature) were selected as the biomarker candidates. In the validation phase, enzyme-linked immunosorbent assays (ELISAs) were applied to measure the plasma levels of the seven candidate proteins from 63 participants: 26 CCA patients, 17 DC, and 20 normal individuals. Four proteins, S100A9, AACT, NGAL, and PSMA3, were significantly increased in the CCA group. To generate the multiplex biomarker assays, nine machine learning models were trained on the plasma dynamics of all seven candidates (All-7 panel) or the four significant markers (Sig-4 panel) from 45 of the 63 participants (70%). The best-performing models were tested on the unseen values from the remaining 18 (30%) of the 63 participants. Very strong predictive performances for CCA diagnosis were obtained from the All-7 panel using a support vector machine with linear classification (AUC = 0.96; 95% CI 0.88–1.00) and the Sig-4 panel using partial least square analysis (AUC = 0.94; 95% CI 0.82–1.00). This study supports the use of the composite plasma biomarkers measured by clinically compatible ELISAs coupled with machine learning models to identify individuals at risk of CCA. The All-7 and Sig-4 assays for CCA diagnosis should be further validated in an independent prospective blinded clinical study.

## 1. Introduction

Cholangiocarcinoma (CCA) is an aggressive malignant tumor found in the epithelial cells lining the biliary tree [1,2,3]. Its prevalence varies worldwide, however, CCA imposes a major public health threat in Southeast Asian countries, particularly Thailand, and is often associated with *Opisthorchis viverrini* (OV) infestation and nitrosamine intake [4,5,6]. The highest incidence of CCA is found in the province of Khon Kaen, Northeast Thailand, where the age-standardized annual incidence rates are 36 per 100,000 in females and 88 per 100,000 in males [1,2]. The worldwide incidence of CCA has been increasing over the past 30–40 years to ~2% of all cancer-related deaths and 18% of all liver cancers. CCAs are divided into three types based on their anatomical localization; (i) intrahepatic CCA, which originates from the small bile ducts, (ii) perihilar CCA, and (iii) distal CCA, which originates from the ductal epithelium of the extrahepatic biliary tree [5,7,8,9,10,11]. The prognosis of patients with CCA is poor because of its initial silent clinical characteristics and its rapid growth and aggressive metastasis in the late stages. Hence, most patients are diagnosed at an advanced stage when treatment is less effective and the prognosis is poor [1,2,8,11,12]. The current diagnosis of CCA requires a combination of clinical, biochemical, radiological, and histological information [7]. Different imaging techniques may be used for the diagnosis of each CCA subtype, such as ultrasonography, computed tomography (CT), percutaneous transhepatic cholangiography, and endoscopic retrograde cholangiopancreatography [7,11]. However, these techniques are not desirable for initial testing due to the cost burden, variable degrees of accuracy, and limited accessibility [11]. Improved detection of this cancer with a simpler and less invasive approach, such as plasma biomarkers, would be of substantial clinical benefit for diagnosis, monitoring, and predicting outcomes for CCA patients [11].

The most widely used clinical biomarkers for CCA diagnosis include carbohydrate antigen 19-9 (CA19-9) and carcinoembryonic antigen (CEA). However, both CA19-9 and CEA are not specific to CCAs; they also increase in many other liver diseases, including alcoholic liver disease, viral hepatitis, primary sclerosing cholangitis (PSC), cholestasis, liver injury, and other cancer types [1,6,9,11,13]. CA19-9 has large variations in sensitivity (50–90%) and specificity (54–98%) and may be elevated in benign biliary disease or cholangitis. For the diagnosis of intrahepatic CCA, the sensitivity and specificity of CA19-9 are 62% and 63%, respectively, while primary sclerosing cholangitis (PSC) patients have 75% sensitivity and 80% specificity in diagnosing extrahepatic CCA by CA19-9 [8]. However, a high CA19-9 level of >1000 U/mL has been associated with metastatic intrahepatic CCA and might be used in disease staging rather than diagnosis [8]. Similarly, the CCA diagnostic sensitivity of CEA ranges from 42% to 85% and CEA specificity ranges from 70% to 89% [7,14,15]. High levels of CEA are often observed in gastrointestinal cancer, especially in colorectal carcinoma, and may also be observed in cholangiocarcinoma [16]. Moreover, the low sensitivity/specificity and poor early detection limit the clinical usefulness of these markers. 

New biomarkers for CCA detection are needed. Mass spectrometry-based proteomics is a powerful tool for biomarker discovery [17]. Several quantitative proteomic studies using different sample types (plasma, bile, urine, extracellular vesicles, and tissues) and various techniques have been used to search for specific CCA biomarkers [5,7,18]. Gene expression profiling and immunohistochemistry comparing CCA tumor tissues with normal liver tissues identified the potential CCA biomarkers ANXA1, ANXA2, SERPINC1, and AMBP [19]. Proteomic screening also found the overexpression of AMBP protein precursors in cholangiocarcinoma tissue [20]. The secretomes of cholangiocarcinoma cell lines specifically express lipocalin-2 (NGAL) and 49 other proteins that are not expressed by hepatocellular carcinoma cells [21]. High levels of proteasome subunit α type-3 (PSMA3) are in the plasma of CCA patients compared to normal individuals and patients with hepatocellular carcinoma [4]. Thus, AMBP, NGAL, and PSMA3 are also promising potential biomarkers for cholangiocarcinoma. 

This study applied a translational proteomic approach to accelerate CCA biomarker discovery, validation, and multiplex assay development. The accessible potential diagnostic protein markers were investigated in the plasma of CCA patients and compared with normal individuals and disease controls, including non-CCA tumors and non-malignant hepatobiliary pathological conditions. Candidate markers were identified from previous studies [4,19,20,21] and by the label-free proteomic quantitation of nine pooled plasma specimens of a discovery cohort (total n = 27; 9 CCA, 9 normal, 9 DC; 3 samples of each group/pool). The candidate biomarkers were validated by clinically compatible ELISA immunoassays in a larger cohort of 63 patients and controls. Machine learning models were trained and tested on ELISA-measured values of the candidate biomarkers to develop predictive models for CCA diagnosis. The workflow of this study is illustrated in Figure 1.

## 2. Results

### 2.1. Discovery of Candidate Biomarkers by Plasma Proteomics 

In the discovery phase, 27 plasma samples from nine CCA patients (CCA group), nine healthy individuals (normal group), and nine patients with non-CCA tumor or hepatobiliary diseases (disease control group) generated three pooled normal (pN), three pooled CCA (pCCA), and three pooled disease control (pDC) samples. The clinical features of the healthy controls, patients with cholangiocarcinoma, and disease control, including gender, age, the definitive diagnosis, and stage of disease, are shown in Table 1. The disease control group comprised patients who presented with clinical features resembling CCA: jaundice, pale stool, cachexia, low-grade fever, and/or ascites/abdominal mass; the diagnosis of CCA was excluded by standard clinical investigations: computed tomography (CT), endoscopic retrograde cholangiopancreatography (ERCP), and/or tissue biopsy. 

The pooled samples were pre-fractionated by a MARS-14 (multi-affinity removal column, human-14) immunodepletion column (to remove 14 highly abundant plasma proteins) before in-solution tryptic digestion and label-free quantitation (LFQ) mass spectrometry (full details in the Methods section). Each pooled sample was analyzed in three technical replicates, resulting in a total of 27 LC-MS/MS runs. A total of 1595 peptides, corresponding to 248 unique proteins, were identified and quantified across 27 injections at a 1% false discovery rate (FDR) using Progenesis label-free LC-MS software v.3.1 (Appendix A contains the full dataset). 

The global proteome profiling of 248 plasma proteins was analyzed using a heatmap with unsupervised clustering (Figure 2a). The hierarchical clustering clearly separated the normal control group from the CCA and disease control groups, even though pCCA 1 (which represented early-stage CCA) showed considerable similarity to the normal control group. Then, differential expression analysis was performed to detect the candidate biomarkers at the thresholds of a 1.5× fold-change and *p* < 0.05, adjusted for the post-hoc analysis of multiple comparisons. Accordingly, 24, 6, and 21 differentially expressed proteins were found in the comparisons of pCCA vs. pN, pCCA vs. pDC, and pDC vs. pN, respectively (Figure 2b). Appendix A lists all significant proteins with their fold changes.

### 2.2. Rationale for Selection of the Candidate CCA Biomarkers 

From our perspective, good candidate biomarkers should be recognized by at least two independent studies for better reproducibility, or one study with a highly confident biomarker potential. For multiplexing biomarkers, each should represent distinct pathogenic conditions or states for better coverage of disease heterogeneity and to maximize specificity for the disease. Accordingly, four significant proteins from our proteomic analysis, S100A9, AACT, AFM, and TAOK3, and three potential CCA biomarkers from previous studies, NGAL, PSMA3, and AMBP, were selected for further validation using the clinically compatible antibody-based assay. It is noteworthy that the label-free proteomic quantitation in our work may be able to detect intermediate-to-low abundance plasma proteins at the concentration range of nanograms per milliliter (33). The use of ELISAs with greater detection sensitivity from low nanograms to picograms per milliliter level (according to the manufacturer’s instructions) would offer a channel to evaluate the potential contributions of previously identified biomarkers, even though they were missed during the proteomic biomarker screening. The specific rationale for the selection of the candidate biomarkers is provided in Table 2. 

### 2.3. Validation of the Candidate Biomarkers by ELISA

The potential clinical applicability of the biomarker candidates was validated in the validation patient cohort of n = 63:26 CCA, 20 normal controls, and 17 disease controls (demographic data in Appendix A). In addition, unlike the discovery plasma proteomics that analyzed the MARS14-immunodepleted plasma, the whole unfractionated plasma was measured by the clinically compatible ELISAs to test the clinical relevance of the identified biomarkers more stringently. Figure 3 shows the plasma levels of S100A9, AACT, AFM, TAOK3, NGAL, PSMA3, and AMBP proteins. The results show that the CCA patients had significantly higher plasma S100A9, AACT, NGAL, and PSMA3 levels relative to the normal controls (Figure 3). The plasma S100A9 and AACT proteins of the CCA group were also significantly higher than those of the DC group (Figure 3). Plasma AFM, AMBP, and TAOK3 were not statistically different among the groups. This finding identifies plasma S100A9, AACT, NGAL, and PSMA3 proteins as the potential biomarkers for CCA diagnosis, although it is unlikely that any protein could serve as the CCA biomarker alone.

### 2.4. Diagnostic Performance of the Multiplex CCA Markers

To address whether these candidate CCA markers could be combined to build a multiplex assay, pairwise scatter plots of all combinations of plasma S100A9, AACT, AFM, TAOK3, NGAL, PSMA3, and AMBP proteins evaluated their composite effects on the separation of the CCA vs. non-CCA (collapsing normal and DC) groups (Figure 4). The data were normalized by log2 transformation to reduce potential biases due to differences in the order of magnitude of the plasma concentrations of the candidate proteins, and the boxplots of the transformed data in Figure 4a are consistent with the non-transformed data in Figure 3. The S100A9, AACT, NGAL, and PSMA3 proteins were increased in the CCA group. The AFM protein trended toward increasing in CCA, while the TAOK3 and AMBP proteins were unchanged among groups. Next, the transformed data were arranged as pairwise scatter plots, resulting in a total of 21 combinations of two candidate biomarkers (Figure 4b). As anticipated, different combinations of the candidate proteins delivered dissimilar patterns between the CCA and non-CCA sample separation, thereby supporting further investigation into using multiple candidate biomarkers for CCA diagnosis. 

Two composite biomarker panels were designed for testing. The All-7 panel consisted of all candidate CCA biomarkers (S100A9, AACT, AFM, TAOK3, NGAL, PSMA3, and AMBP) identified by the plasma proteomic analysis of the nine pooled samples and from the literature (Figure 2 and Table 2). The Sig-4 panel consisted of four proteins (S100A9, AACT, NGAL, and PSMA3) that were successfully validated by ELISA (Figure 3). 

Machine learning classification was coupled with the multiplex biomarker assays, aiming for the improvement of their diagnostic performance. Nine machine learning models, including the Bayesian generalized linear model (bayesglm), generalized linear model (glm), k-nearest neighbors (knn), naïve Bayes (nb), neural network (nnet), partial least squares (pls), random forest with 1000 decision trees (rf1000), support vector machine (SVM) with linear classification (svm_linear), and SVM with radial kernel function (svm_radial) were trained on the training dataset (*n* = 45/63 (70%), using 19 CCA vs. 26 non-CCA (17 normal and 9 DC; Appendix A) with 10-fold cross-validation (details of parameter tunings in Appendix A). The results indicate that the svm_linear model exhibited the best ranking with the area under the receiver operating characteristic curve for the All-7 panel, and the pls exhibited the best ranking for the Sig-4 panel. The diagnostic performances of the All-7 with the svm_linear model and the Sig-4 with the pls model were validated using the unseen testing dataset (*n* = 18/63 (30%); 7 CCA vs. 11 non-CCA, 3 normal and 8 DC; Appendix A). The receiver operating characteristics show strong predictive performances for CCA diagnosis using the svm_linear model on the All-7 panel (AUC of 0.961; 95% CI of 0.885-1.000) and the pls model on the Sig-4 panel (AUC of 0.935; 95% CI of 0.819–1.000) (Figure 5b). The predictive performances of all nine models for the All-7 and the Sig-4 panels are shown in Appendix A.

## 3. Discussion

This study applied translational research principles by identifying the candidate CCA biomarkers in a small number of patients, validating their potential usefulness in a larger patient cohort, and developing multiplex biomarker predictive models that warrant further prospective diagnostic studies. Lessons learned in the past suggest that it is unlikely to discover a single novel plasma protein with exceptional cancer diagnostic performance [4,5,7,18,21,33]. Instead, the combination of multiple plasma proteins associated with different aspects of CCA heterogeneity may allow for the identification of CCA patients at various disease stages. 

To achieve this goal, the identification and selection of the CCA biomarker candidates did not rely solely on high-throughput proteomic analysis (Figure 2) but also took into account the feasibility of the identified proteins in relation to previous independent studies of CCA biomarkers [4,6,19,21,22,23,24,25,26,27,29,30,31,32], thereby reflecting several CCA types and pathogenic conditions (Table 2). ELISA, a clinically compatible antibody-based assay, was chosen for biomarker validation. Finding increased levels of plasma S100A9, AACT, NGAL, and PSMA3 proteins in 26 CCA patients relative to 20 normal individuals and 17 patients with non-CCA diseases allowed us to develop the All-7 and the Sig-4 panels. Multiplex assays (Figure 3) provided a database from which machine learning developed the predictive models using the training dataset. The most promising models exhibited strong diagnostic performances (AUC > 0.9) when coupled with the All-7 and the Sig-4 panels (Figure 5). Nonetheless, the true diagnostic performance of All-7 ELISAs vs. Sig-4 ELISAs, in conjunction with their time- and cost-effectiveness, require testing in independent clinical trials. 

Although this translational proteomic project delivered potentially useful multiplex assays for CCA diagnosis, several limitations remain to be addressed: 

Firstly, the relatively small sample size of this study in the discovery (nine pooled samples of 27 individuals) and validation cohorts (63 individuals) allow the possibility that the biomarkers discovered and validated may not generalize to larger cohorts due to unknown variations of the measured biomarkers at the population level [34,35]. To address this issue, this study developed multiplex panels including only biomarkers with previous evidence of positive outcomes in several independent cohorts (Table 2), implying that the developed multiplex assays could be applied to many, if not all, populations of CCA. Nonetheless, the true diagnostic performance of the All-7 and Sig-4 panels in the general population requires further validation. 

Secondly, during the proteomic discovery phase, this study may have missed some novel (and valid) biomarkers due to the selection process that prioritized reproducibility over novelty. For example, APC membrane recruitment protein 1 (AMER1) significantly increased in the pCCA compared to the pDC and pN groups (Figure 2). Nonetheless, the AMER1 protein has never been studied in cholangiocarcinoma and thus was not prioritized for further validation in this study. Follow-up studies may consider including more proteins of interest for the validation phase to potentially strengthen the final assay. 

Lastly, this study developed the multiplex biomarker assays coupled to the top-performance trained models, which showed strong predictive performance for detecting CCA with the AUC > 0.9. Nevertheless, this result is based on a single machine learning model. The ensemble-based machine learning method could possibly exhibit better performance, stability, and predictive accuracy [36]. Future studies of CCA biomarkers should be pursued in prospective multicenter or population-based cohorts. Bile analysis for proteins, as well as the correlation between the measured biomarkers and the CCA stages, should also be included. Additional biomarkers of interest may be added to the All-7 or Sig-4 panels coupled with the ensemble-based machine learning method, aiming to maximize the diagnostic accuracy of early CCA. 

Nonetheless, the current study strongly supports the utility of the described novel approach toward identifying candidates for use in building more sophisticated biomarker assays: identification of a panel of relevant biomarker proteins; testing potential biomarkers by ELISA; in silico identification of the most potent biomarker combinations; and in silico machine learning to identify the panel of biomarkers and the program for processing clinical data. The resultant final assay holds great promise for earlier and more precise detection of life-threatening diseases.

## 4. Materials and Methods

### 4.1. Plasma Collection

EDTA-blood tubes were collected at Sappasitthiprasong Hospital, Ubon Ratchathani, Thailand, as left-over specimens. Healthy individuals who presented at the hospital for an annual check-up without a history of underlying disease comprised the normal controls. Definitive diagnosis for individuals with intrahepatic, perihilar, or distal CCA identified the CCA group. Diagnoses of underlying hepatobiliary diseases (disease control; DC) were made based on the histopathological examination of biopsy or surgical specimens. The EDTA-blood was centrifuged at 380× *g* for 15 min at 4 °C to obtain plasma specimens, which were aliquoted and stored at −80 °C until use. The study was approved by the local Ethics Committee of the Faculty of Medicine, Ramathibodi Hospital, Mahidol University and Sappasitthiprasong Hospital (protocol ID 03-58-68; approved on 8 May 2015; last amended on 4 May 2018). Written informed consent was waived due to the use of discarded de-identified specimens. 

### 4.2. Immunodepletion of High Abundance Plasma Proteins 

MARS-14 columns (4.6 × 100 mm), purchased from Agilent Technologies, Inc., were used to deplete the 14 most abundant proteins (albumin, immunoglobulin gamma (IgG), antitrypsin, IgA, transferrin, haptoglobin, fibrinogen, alpha2-macroglobulin, alpha1-acid glycoprotein, IgM, apolipoprotein AI, apolipoprotein AII, complement C3, and transthyretin) from the pooled plasma samples. Immunodepletion was performed at room temperature using an Agilent 1260 Infinity high-performance liquid chromatography (HPLC) system. Briefly, the MARS-14 column was injected with 80 µL of the diluted plasma (1:3 plasma/buffer A) at a low flow rate (0.125 mL/min) for 18 min and then at a flow rate of 1 mL/min for 2 min. The flow-through fraction (representing the depleted plasma) was collected. For reusing the column, the system was changed to 100% buffer B (elution buffer), to elute the bound proteins at a flow rate of 1 mL/min for 7 min. The column was then regenerated by equilibration in 100% buffer A for 11 min at a flow rate of 1.0 mL/min. The detector was set at a wavelength of 280 nm. The flow-through fractions were pooled and concentrated using a Spin-X UF 500 concentrator (5 kDa MW cut-off; Corning Life Sciences, Tewksbury, MA, USA) centrifuge containing a fixed-angle rotor at 15,000× *g* for 30 min at 4 °C. The protein concentration was estimated using the Bradford assay. 

### 4.3. In-Solution Tryptic Digestion

Ten micrograms of protein were reduced with 100 mM DTT (10 mM final concentration) for 5 min at 95 °C. Alkylation was performed using a 1/10 volume of 200 mM iodoacetamide and incubated for 30 min at room temperature in the dark. The proteins were then digested by a 1:50 (*w*/*w*) sequencing grade trypsin (Promega Corporation, Madison, WI, USA) at 37 °C overnight. The digestion reaction was stopped by adding formic acid to reach a 1% final concentration, and the samples were evaporated to dryness in a SpeedVac. The samples were purified by C18 ZipTip® (MilliporeSigma, Burlington, MA, USA) and stored at −20 °C until they were used for analysis.

### 4.4. Label-Free Quantitation Mass Spectrometry 

The digested samples were dissolved in 0.1% formic acid in water. Each pooled plasma sample was run in triplicate in a nano-flow liquid chromatography system (Thermo Fisher Scientific, Inc., Waltham, MA, USA) coupled with the amaZon speed ion trap mass spectrometer (Bruker Corporation, Billerica, MA, USA). A C18 Acclaim PepMap RSLC (75 µm i.d. × 150 mm) column (Thermo Fisher Scientific, Inc.) was used to desalt and concentrate tryptic peptides. An LC gradient of 1–50%B for 70 min, 50–90%B for 5 min, followed by 90%B for 15 min was obtained by combining mobile phase A (0.1% formic acid in water) and mobile phase B (0.1% formic acid in 100% ACN). One microliter of the sample containing 100 ng/µL was injected into the nano-LC system prior to separation by the gradient.

Progenesis label-free LC-MS software (version 3.1; Nonlinear Dynamics, Newcastle upon Tyne, UK) was used to identify and quantify peaks in the raw data from the LC-MS/MS. Data alignment was based on the LC retention time of each sample. A reference sample was established, the retention times of all other replicates were aligned to this reference, and the peak intensities were then normalized. Data from the MS/MS spectra were searched using Mascot software version 2.4.0 (www.matrixscience.com, accessed on 13 August 2022) against the SwissProt (Homo sapiens) database. The following search parameters were used for protein identification: MS/MS mass tolerance set to 0.6 Da; peptide mass tolerance set to 1.2 Da; carbamidomethylation set as a fixed modification; mass peaks (features) with charge states +2, +3, and +4; ESI-TRAP instrument; and ≤1 missed cleavages were allowed. Significant peptide identifications above the identity or homology threshold were adjusted to a ≤1% peptide false discovery rate (FDR) using the Mascot Percolator algorithm. Peptides were considered valid if their Mascot ion score was over 30. After the spectral counts were normalized, comparisons of each protein expression were performed. 

### 4.5. ELISA

Commercially available ELISA kits were used to measure the plasma concentrations of the biomarker candidates in the cohort of 63 (26 CCA, 20 normal controls, 17 DC). The ELISA kits included: S100A9 (E-EL-H1290; Elabscience, Wuhan, China), AACT (ab157706; Abcam, Cambridge, UK), AFM (MBS2704330; MyBioSource, Inc.), TAOK3 (KTE60470; Abbkine, Inc, Wuhan, China), NGAL (BMS2202; eBioScience, Vienna, Austria), PSMA3 (MBS9336584; MyBioSource, Inc., San Diego, CA, USA), and AMBP (MBS564034; MyBioSource, Inc.). All assays were performed on whole plasma according to the manufacturers’ instructions. The optical density (OD) was measured on a SpectraMax M2 Microplate Reader (Molecular Devices) at 450 nm.

### 4.6. Data and Statistical Analyses

Data and statistical analyses were performed using Excel and R programs. Multiple comparisons were performed by one-way analysis of variance (ANOVA) with Tukey’s post-test or Wilcoxon’s signed-rank test, as appropriate. Proteomic data analysis and visualization were performed using our custom bioinformatic workflow as described previously [37]. Machine learning was performed using caret, ranger, and arm packages. Data preprocessing was performed by log2 transformation followed by a 70:30 splitting assigned to the training and testing datasets. The data were centered and scaled, and then 10-fold cross-validation was performed to fit the training model. Receiver operating characteristics (ROCs) were used to determine the predictive performance of the trained model on the unseen testing dataset, where a 95% confidence interval (CI) of the area under the curve (AUC) was calculated by the DeLong method. *p*-values < 0.05 were considered statistically significant.

## 5. Conclusions

This report describes a translational proteomic approach for the identification of CCA, including biomarker discovery by high-throughput proteomic analysis, biomarker validation by clinically compatible immunoassays, and multiplex assay generation with the support of machine learning models. The performance of the All-7 and the Sig-4 multiplex assays can now be further validated in a full-sized clinical prospective cohort or multicenter study. When fully validated, this assay holds great promise for earlier and more precise detection of cholangiocarcinoma. Moreover, this novel approach to developing multi-biomarker multiplex assays may be used as a general strategy to address many other dire diseases. 

## Figures and Tables

**Figure 1 molecules-27-05904-f001:**
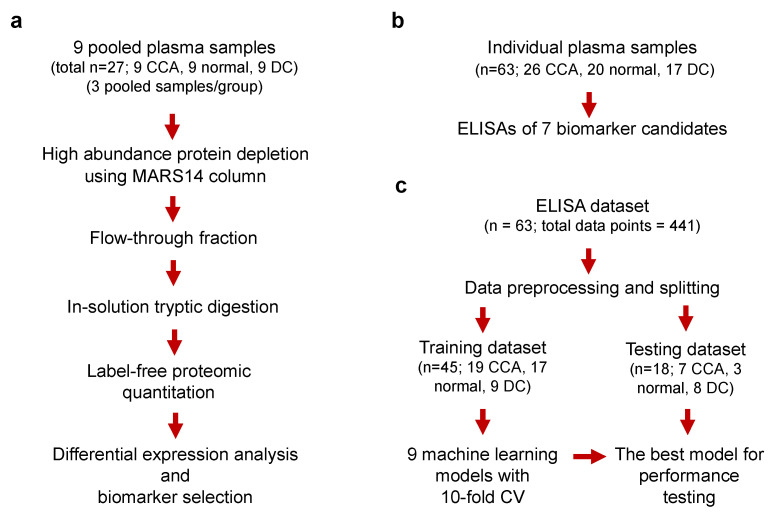
The workflow of this study. (**a**) Biomarker discovery by proteomics. (**b**) Biomarker validation by enzyme-linked immunosorbent assays (ELISAs). (**c**) Multiple assay generation by machine learning. CCA, cholangiocarcinoma; CV, cross-validation; MARS-14, multi-affinity removal column, human-14; DC, disease control.

**Figure 2 molecules-27-05904-f002:**
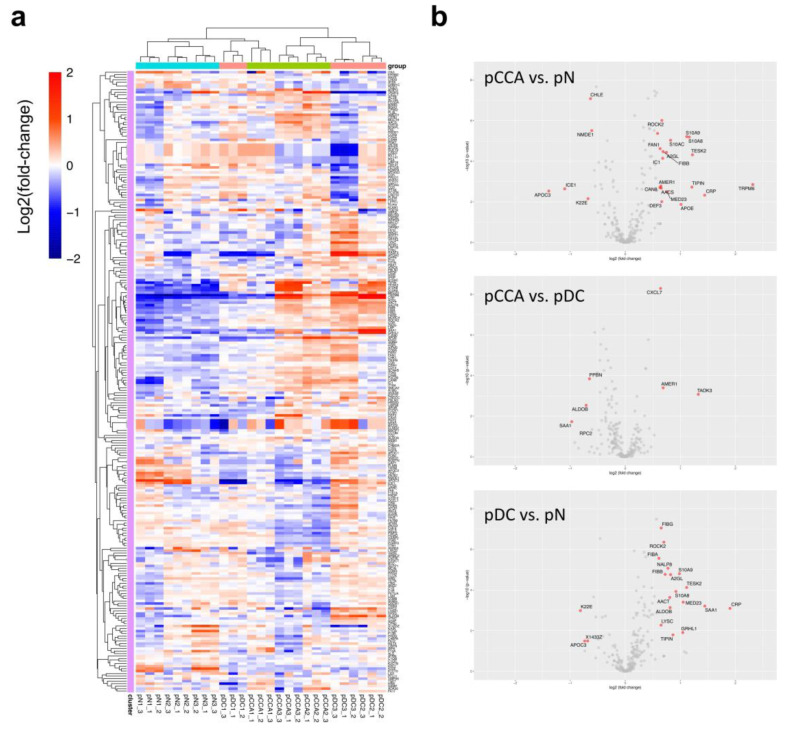
Cholangiocarcinoma biomarker discovery by plasma proteomic analysis. (**a**) Heatmap with unsupervised hierarchical clustering of 248 protein expressions across 27 injections corresponding to nine pooled plasma samples with three technical replicates (details in Table 1 and Appendix A). (**b**) Differential expression analysis of pooled plasma samples of the normal control (pN), CCA (pCCA), and disease control (pDC) groups. The proteins are shown in rows and the samples are arrayed by column. Red indicates upregulation and blue indicates downregulation relative to the median expression (white) of each protein across all samples. (**b**) Volcano plot demonstrates the significant proteins (red color) at the thresholds of a 1.5× fold-change and *p* < 0.05 after multiple comparisons using ANOVA with Tukey’s post-hoc analysis. X-axis is log2 fold change. Y-axis indicates −log10 (*p*-value).

**Figure 3 molecules-27-05904-f003:**
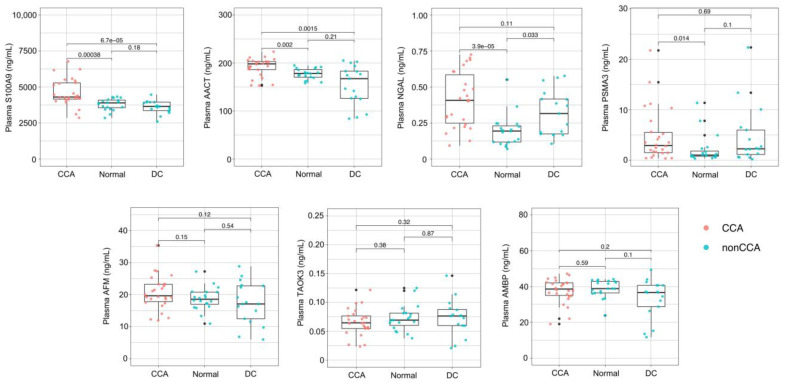
ELISA-based biomarker validation. Levels of seven candidate biomarkers were determined in the whole plasma of CCA (*n* = 26), normal controls (*n* = 20), and disease control (DC; *n* = 17). AACT, alpha-1-antichymotrypsin; AFM, afamin; AMBP, alpha-1 microglobulin; NGAL, neutrophil gelatinase-associated lipocalin; PSMA3, proteasome subunit alpha type-3; TAOK3, TAO kinase 3.

**Figure 4 molecules-27-05904-f004:**
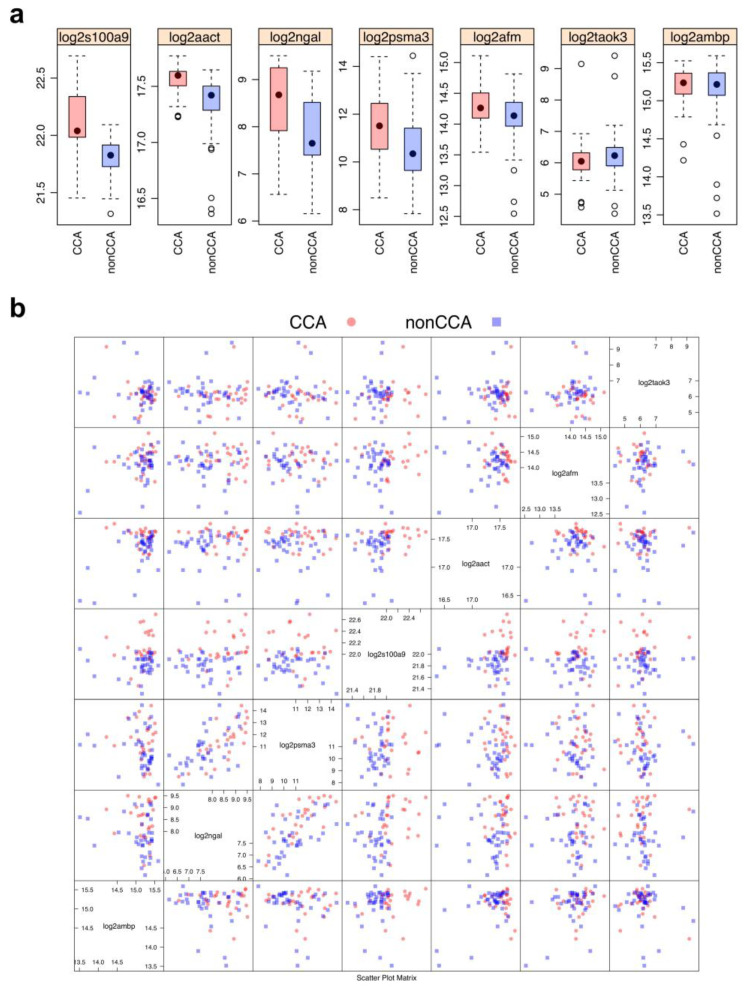
Data pre-processing and exploration prior to machine learning modeling. (**a**) Boxplots show the log2-transformed intensity of seven candidate biomarkers of CCA (*n* = 26) vs. non-CCA (*n* = 37; 17 disease controls, 20 healthy individuals). (**b**) Pairwise scatter plots of combined CCA vs. non-CCA potential biomarkers.

**Figure 5 molecules-27-05904-f005:**
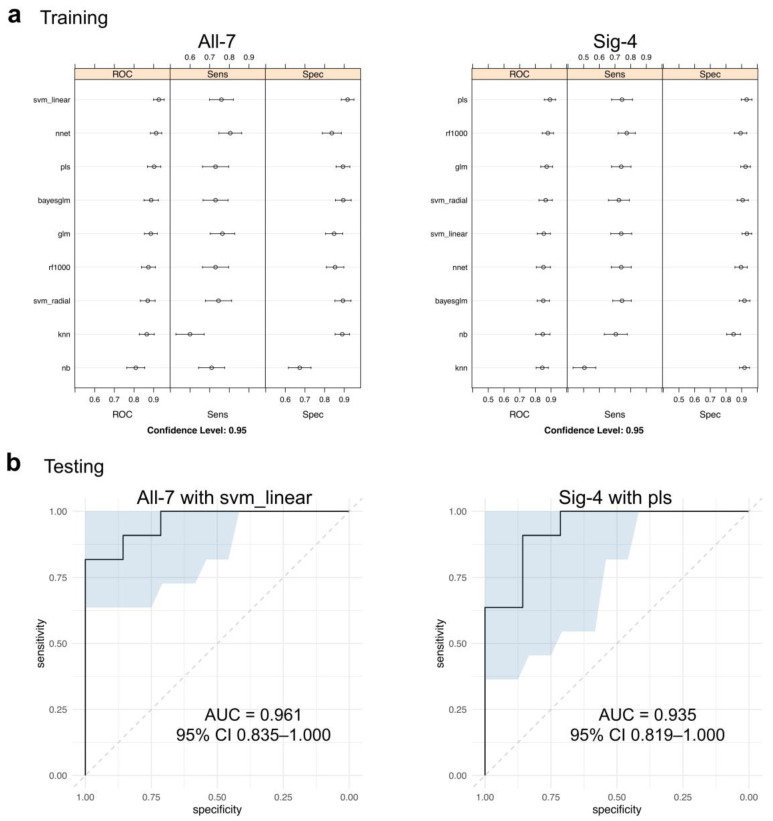
Generation of multiplex CCA biomarker assays coupled to machine learning classification. (**a**) The performance of nine models trained on a 70% subset of the parent dataset (*n* = 45) with 10-fold cross-validation. Ranking by the area under the ROC curve, the support vector machine with linear classification (svm_linear) was the best performing model for the All-7 panel, and partial least square (pls) performed best for the Sig-4 multiplex biomarker panels. (**b**) CCA diagnostic performances of the All-7 with svm_linear and the Sig-4 with pls against the unseen testing dataset (*n* = 18). Abbreviations: bayesglm, Bayesian generalized linear model; glm, generalized linear model; knn, k-nearest neighbors; nnet, neural network; nb, naïve Bayes; pls, partial least squares; rf1000, random forest with 1000 decision trees; ROC, receiver operating characteristics; Sens, sensitivity; Spec, specificity; svm_linear, support vector machine with linear classification; svm_radial, support vector machine with radial kernel function.

**Table 1 molecules-27-05904-t001:** Characteristics of CCA, normal, and disease control samples.

Pooled Sample	Gender	Age	Condition/Disease	CCA Stage
	M	46	Cholangiocarcinoma, perihilar	I
pCCA1	M	51	Cholangiocarcinoma, distal	IIa
	M	73	Cholangiocarcinoma, distal	IIb
	M	67	Cholangiocarcinoma, intrahepatic	III
pCCA2	F	55	Cholangiocarcinoma, intrahepatic	IIIA
	F	46	Cholangiocarcinoma, intrahepatic	IIIA
	M	50	Cholangiocarcinoma, metastasis	IV
pCCA3	M	55	Cholangiocarcinoma, intrahepatic	IV
	F	51	Cholangiocarcinoma, intrahepatic	IV
	F	51	Healthy	-
pN1	M	56	Healthy	-
	F	52	Healthy	-
	M	55	Healthy	-
pN2	F	59	Healthy	-
	F	50	Healthy	-
	M	54	Healthy	-
pN3	F	56	Healthy	-
	M	66	Healthy	-
	F	72	HCC, chronic cholecystitis	-
pDC1	M	52	HCC, cirrhosis	-
	F	61	HCC	-
	M	34	Chronic HBV infection	-
pDC2	F	64	Chronic cholecystitis, DM, HT	-
	M	56	Periductal fibrosis	-
	F	33	Focal nodular hyperplasia, liver	-
pDC3	M	59	Granulomatous inflammation, CBD	-
	F	64	Gastrointestinal stromal tumor	-

Abbreviations: CBD, common bile duct; DM, diabetes mellitus; F, female; HBV, hepatitis B virus; HCC, hepatocellular carcinoma; HT, hypertension; M, male; pCCA, pooled cholangiocarcinoma sample; pDC, pooled disease control sample; pN, pooled normal sample.

**Table 2 molecules-27-05904-t002:** The selected candidate CCA biomarkers for the validation study.

Gene Name	Accession	Protein Name	Rationale for Selection	Reference
*S100A9*	S10A9_HUMAN	Protein S100-A9	Significantly upregulated in pCCA vs. pN and pDC (*p* < 0.001) Previously identified as a CCA biomarker in multiple independent studies	This study [22,23,24,25]
*SERPINA3*	AACT_HUMAN	Alpha-1-antichymotrypsin	Significantly upregulated in pCCA vs. pN and pDC (*p* < 0.001) Previously proposed as a candidate biomarker of opisthorchiasis-associated CCA	This study [26,27]
*AFM*	AFAM_HUMAN	Afamin	Significantly downregulated in pCCA vs. pN (*p* < 0.001) Previously identified as a biomarker of advanced CCA with poor prognosis	This study [28,29]
*TAOK3*	TAOK3_HUMAN	Serine/threonine-protein kinase TAO3	Significantly upregulated in pCCA vs. pDC (*p* < 0.001) A tumor suppressor gene with genomic evidence of significant alteration in CCA	This study [30]
*NGAL*	NGAL_HUMAN	Neutrophil gelatinase-associated lipocalin	Previously identified as a biomarker of perihilar CCA, which could distinguish CCA from benign biliary tract diseases	[21,31,32]
*PSMA3*	PSA3_HUMAN	Proteasome subunit alpha type 3	Previously identified as a CCA biomarker from the CCA cell secretome and successfully validated using an antibody-based assay in 12 clinical plasma samples (5 normal, 4 CCA, 3 DC)	[4] ^a^
*AMBP*	AMBP_HUMAN	Alpha-1 microglobulin	Previously identified as a biomarker of intrahepatic CCA	[19,20]

**^a^** PSMA3 is justified by our previous study as having highly confident CCA biomarker potential.

## Data Availability

All data are available in the manuscript and the Appendix A. The trained machine learning models are available from the corresponding author (S.C.) upon reasonable collaborative request.

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
