# Peer review of "Translational Proteomic Approach for Cholangiocarcinoma Biomarker Discovery, Validation, and Multiplex Assay Development: A Pilot Study"

_molecules, 2022, doi:10.3390/molecules27185904_

Round 1

Reviewer 1 Report

1- I agree with the authors about the small number of subjects included in the study. So, I suggest for them to add the two words "A pilot study" at the end of title of the manuscript. 

2- In the material and method, it is not clear for cholangiocarcinoma whether it is intrahepatic or extrahepatic?

3- The cost effectiveness of the  7 and 4 multiplex assays needs more discussion.

4- Is it possible to discuss the proteomic markers discovered in the light of liver enzymes of CCA

5- Prospectively, the study should recommend bile analysis for the proteins as well as the correlation between the protein markers detected and stages of the CCA 

Reviewer 2 Report

The paper “Translational Proteomic Approach for Cholangiocarcinoma: Biomarker Discovery, Validation, and Multiplex Assay Development by K Watcharatanyatip et al is a interesting study and can contribute to face difficulties related to early detection on CCA. The study has been well designed and results support the preliminary hypothesis. In fact, prevention and early diagnosis remain the cornerstone for improving the survival of this devasting disease. Furthermore, identifying preventable risk factors and patients at risk of CCA are keys to decreasing the disease-related mortality of this highly lethal neoplasia. 

Surveillance should be considered for people at risk as those affected by Primary Sclerosing Cholangitis that is recognized as the most important risk factor for CCA development in Western countries. Imaging techniques  such as US, CT, MRI and MRCP can be useful for early detection of CCA onset in PSC patients. US specificity and sensitivity are low, CT and MRCP are invasive tools in terms of exposure to high dose of radiation. MRI should be considered as the study of choice as it is superior against US in the detection of early-stage perihilar CCA in patients with PSC, showing better area under the curve (AUC) in the entire cohort (0.87 vs. 0.70) and also in asymptomatic patients (0.81 vs. 0.59). Anyway, the use of MRI as screening test is higly expensive for Health Services. The gold standard may be the use of biomarkers for screening due to their worldwide availability and reproducibility. Taking this in mind authors have contributed to this aim with their study.

The application of machine learning models is quite promising. I’m not an expert but I’m strongly beleive in their role in improving early detection of CCA so as AI can do in extractiong an increased number of featrures from Imaging techniques. The combination of multiple plasma proteins associated with different aspects of CCA heterogeneity may together with allow identification of CCA patients at various disease stages.

I also very much appreciated the author's correctness in clearly explaining the limitations of this study.

Some open questions:

1.   Surveillance in cancer is defined as the repeated application of a test over time with the aim of reducing mortality from a disease. It is critical to tease apart between mortality (measured as the number of deaths per unit of time) and survival (duration of life after the diagnosis of the disease). Decreasing cancer-related mortality should be the sole objective of surveillance programs since survival is a surrogate endpoint that is subject to multiple biases that do not impact mortality. The ability of surveillance to detect the disease at an earlier stage is a required outcome of surveillance programs, but the mere finding of early-stage disease is not sufficient as proof of efficacy. In addition, treatment at an early stage should impact on survival in most patients. Otherwise, its effect on mortality will not be evident.

Can the authors can provide more information on whether their study may have an impact in terms of OS or PFS, based on a quantitative or qualitative analysis of upregulated proteins.

2.   In figure 2b authors report in the Volcano’s plot the significant red protein overexpressed in CCA with respect to DC and N samples. In these plots (Does S10A9 in the first PCCA vs PN graph stand for S100A9?) only some proteins of the All-7 panel can be observed. The lacking proteins are then downregulated or what else?

Authors are asked to clarify

3.   Authors report in the result section: “Plasma AFM levels had a trend of being increased in the CCA group compared to both controls, even though proteomics suggested its downregulation. May this result respresent a bias due this contradictory indication for further clinical application of the set of biomarkers.

Authors are asked to clarify

4. Authors considered nine machine learning models. Among them only two the svm linear model for the ALL-7 panel and pls model for the Sig-4 panel have given satisfying outcomes.

Authors are requested to discuss this result

Minor revision

1.      Figure 1: The I has to be substituted by 1

Reviewer 3 Report

The manuscript by Wactharatanyatip et al describes the proteomic analysis of a small pool of CCA patients along with healthy and disease controls. Potential biomarkers are chosen from this data, informed by previous literature with some candidate markers chosen solely based on existing literature. Validation of the candidates is carried out by ELISA and the data examined to evaluate if multiplexed markers would be a viable strategy for CCA diagnosis.

The paper is overall well written, the disease background and importance and incidence rate are clearly described. The methods and rationale for experimental design are generally well described and the limitations of the study are clearly acknowledged.

However there are some issues with the manuscript which need to be addressed prior to publication.

The major point is that it is not clear how comparable the literature studies used to choose the candidate biomarkers were to the present study. This needs more detail and rationale.

That several biomarkers are literature refs only implies that they were either not found, or not found to significantly differ in the proteomic samples of the study. This needs to be addressed, and is in my view a major limitation of the manuscript in the present format.

Also, the authors do give a rationale for not following up new markers found in this study they should still be reported as this will add to the existing literature. Also it seems contradictory to the inclusion of PSMA3 which only appeared in one it study and was not confirmed in the authors study.

At the end of the paper I am left unsure as to the value of the proteomic screen, would the authors have conducted the same ELISA and reached the same conclusions without this step? 

Also how many of the candidates already had ELISA data and how consistent was this with the data in the manuscript?

Minor points:

The pooling of candidates is referred to several times in the manuscript in an ambiguous way before it is explained. This should be moved forward to the first time it is mentioned.

As above the disease control is mentioned several times before it is fully defined.

The abstract implies the candidate proteins are chosen from the proteomics, that they are heavily informed by the literature needs to be included.

Discussion of timeframes and/or budget constraints are not appropriate. The study an be simply presented as a pilot study to test the hypothesis/workflow.

The authors mention no desalting/clean-up step between tryptic digest and loading onto LC-MS/MS. If a clean-up step was conducted please add this to the materials and methods section.

Round 2

Reviewer 2 Report

The authors have satisfactorily answered the open questions allowing the text to be more easily understood even by researchers who are not completely experts in the field. The work in the present version can be accepted for publication in Molecules J

Reviewer 3 Report

The authors have overall engaged positively with the comments. They have addressed all of my concerns with the exception of point 2-which is not addressed in the manuscript. 1-2 sentences explaining the rationale for testing biomarkers which are literature refs only would significantly strengthen the narrative of the manuscript. Comparing ELISA sensitivity to their chosen label free proteomic technique with appropriate references would suffice here.
